

# The current landscape of m6A modification in urological cancers

Yaohui Zeng, Cai Lv, Bangbei Wan and Binghao Gong

Department of Urology, Central South University Xiangya School of Medicine Affiliated Haikou Hospital, Haikou, China

## ABSTRACT

N6-methyladenosine (m6A) methylation is a dynamic and reversible procession of epigenetic modifications. It is increasingly recognized that m6A modification has been involved in the tumorigenesis, development, and progression of urological tumors. Emerging research explored the role of m6A modification in urological cancer. In this review, we will summarize the relationship between m6A modification, renal cell carcinoma, bladder cancer, and prostate cancer, and discover the biological function of m6A regulators in tumor cells. We will also discuss the possible mechanism and future application value used as a potential biomarker or therapeutic target to benefit patients with urological cancers.

## INTRODUCTION

More and more studies have shown that epigenetic modifications play an important role in the occurrence and development of a variety of tumors. Many scholars and experts focus on epigenetic modifications to develop potential future approaches for cancer therapies now. There are various carriers to reflect epigenetic information, such as DNA methylation, noncoding RNAs, and histone modifications (*Cavalli & Heard, 2019*). RNA modification is the one of important modification methods which can influence the cell phenotype. It has been demonstrated that RNA modification was associated with embryonic development, cardiovascular diseases (*Wu et al., 2021b*), mitochondrial diseases, neurological disorders (*Suzuki, 2021*), and human cancer (*Barbieri & Kouzarides, 2020*; *Song et al., 2021*), especially human cancer. To this day, it is reported that over 300 types of post-transcriptional modifications could be found in cells (*Boccaletto et al., 2022*), such as N1-methyladenosine, 5-methylcytosine, N6-methyladenosine (m6A), N7-methylguanosine, RNA cap methylations, pseudouridine (*Barbieri & Kouzarides, 2020*). It was validated that the most commonly occurring post-transcriptional modifications in eukaryotic cell was m6A (*Dubin & Taylor, 1975*; *Roundtree et al., 2017*).

M6A was found in mRNA in 1974 and was considered to be associated with the selective processing of mRNA sequences (*Desrosiers, Friderici & Rottman, 1974*). The distribution of m6A modification was determined in 2012 (*Dominissini et al., 2012*). M6A modification is a reversible process and has been found in mRNA, miRNA, lncRNA, and circRNA (*Ma et al., 2019*). The m6A peaks were mainly found in the location near the 5′ untranslated region

Corresponding author
Cai Lv, lvcai815@163.com

(UTR), 3′ UTR, transcription starting site region (TSS), coding sequence (CDS), and the stop codon (*Dominissini et al., 2012*; *Chen et al., 2020c*; *Gan et al., 2021*). It is modulated by kinds of m6A RNA methylation regulators. Writers, a kind of methyltransferases, mediate the methylation and upregulate the level of RNA modification (*Zaccara, Ries & Jaffrey, 2019*). Methyltransferase-like 3 (METTL3), Methyltransferase-like 14 (METTL14), and Wilms tumor 1-associated protein (WTAP) are the three most important writers to form the WMM (WTAP, METTL3 and METTL14) complex and regulate m6A modification (*Ping et al., 2014*). Erasers are in charge of promoting demethylase and remove m6A modification, including fat mass and obesity-associated (FTO) and AlkB homologue 5 (ALKBH5) (*Shi, Wei & He, 2019*). Readers are m6A-binding proteins which are the most abundant and complex. They are related to decoding the information of m6A modification and activating the downstream gene. There are many common proteins used to recognize m6A to function such as YTH N6-methyladenosine RNA binding protein 1-3 (YTHDF1-3), YTH domain containing 1-2 (YTHDC1-2), Insulin-like growth factor 2 mRNA binding protein 1-3 (IGF2BP1-3), Heterogeneous nuclear ribonucleoprotein A2/B1 (HNRNPA2B1) (*Fang et al., 2022*). In the process of m6A modification, the three play their respective roles and regulate the level of m6A modification, thus affecting the cell phenotype.

A great deal of biological processes were regulated by m6A modification. Some m6A RNA methylation regulators were related to RNA regulation and metabolism (*Zhang et al., 2021d*). They acted on RNA splicing (*Deng et al., 2018*), stability, translation (*Frye et al., 2018*), nuclear transfer, degradation (*Wang et al., 2015*), and regulate the interaction between RNA and protein (*Lin et al., 2016*), thus participating in formation and development of various tumors.

As a promising potential tumor marker, M6A is widely used in various tumor disease research, and its close association with the clinical characteristics and prognosis of tumor patients indicates that it may be used as a therapeutic target in future tumor treatment research (*An & Duan, 2022*). Recently, some studies have shown that m6A was associated with the oncogenesis and progress of the urinary system. Renal cell carcinoma (*Zhang et al., 2022*), bladder cancer (*Liu, 2021*) and prostate cancer (*Wan et al., 2022*), among the three most common tumors of the urinary system, have been proven to be closely related to m6A modification.

The purpose of this review is to discuss the relationship between urinary system tumors and m6A modification and various m6A regulators, analyze the possible mechanism of action and the role played by m6A. Its clinical application value in the diagnosis of disease and selecting and conducting the treatment of urological tumors was also explored in this review.

## Survey methodology

We used the keywords 'm6A', 'N6-methyladenosine', 'm6A modification', 'm6A regulators', 'Urologic Neoplasms[Mesh]', 'Carcinoma, Renal Cell[Mesh]', 'Urinary Bladder Neoplasms[Mesh]', 'Prostatic Neoplasms[Mesh]', 'Carcinoma, Transitional Cell[Mesh]', 'Urethral Neoplasms[Mesh]' to conduct a literature search of PubMed database. The

articles which were not associated with m6A modification and urological cancers were excluded.

## M6A and urogenital cancer

More and more studies focused on the relationship between m6A modification and urogenital cancer. Here we aim to summarize the variation of m6A modification and its regulators and consequent influence on tumor cells' malignant phenotype.

## Renal cell carcinoma

Renal cell carcinoma (RCC) is one of the common types of urogenital cancer. The most common types of RCC include clear cell renal cell carcinoma (ccRCC), papillary renal cell carcinoma (pRCC), and chromophobe renal cell carcinoma (chRCC). According to GLOBOCAN data, renal tumors account for about 2.2% of estimated cases and deaths worldwide in 2020 (*Sung et al., 2021*). The early diagnosis and treatment of RCC is always a research hotspot. It was found that the m6A content of total RNA increased in the RCC tumor tissues (*Gan et al., 2021*). A study also found that the m6A level decreased in ccRCC (*Xu et al., 2022b*). In terms of "writers", The expression of METTL3 (*Zhu et al., 2022*), WTAP (*Ying et al., 2021*; *He et al., 2021*) was significantly increased in RCC tissues. The level of METTL14 (*Zhang et al., 2021a*; *Liu et al., 2022d*) and eukaryotic initiation factor 3 (eIF3A) (*Zhang et al., 2021b*) in RCC was significantly decreased. The FTO expression was different in various researches, some results show it was upgraded in ccRCC tumors (*Xiao et al., 2020*; *Shen et al., 2022*). However, others find that the expression of FTO decreased (*Zhuang et al., 2019*; *Strick et al., 2020*). The other "eraser" ALKBH5 downregulated in RCC samples (*Strick et al., 2020*). When we talk about "readers", the quantity of heterogeneous nuclear ribonucleoprotein C (HNRNPC), HNRNPA2B1, YTHDC1 (*Von Hagen et al., 2021*), YTHDF1-3 (*Mu et al., 2020*; *Xu et al., 2022a*), and IGF2BP2 (*Li et al., 2022*) was significantly lower than adjacent normal tissues. More detailed information of the variation of m6A regulators in RCC can be found in Table 1. PRCC is the second most common type of renal carcinoma, different types with different m6A modulators expressions. It was reported that the expression levels of HNRNPC and YTHDF1 increased, while the expression levels of Vir-like m6A methyltransferase associated (VIRMA/KIAA1429), zinc finger CCCH-type containing 13 (ZC3H13), METTL14, ALKBH5, and YTHDF2 decreased in pRCC tissues (*Yang et al., 2021b*). *Lei et al. (2022)* revealed that the transcription level of WTAP reduced in pRCC and chRCC. Different m6A RNA methylation regulators also make different contributions to RCC. Some had a potential antitumor function in RCC. METTL14 knockout significantly increased the migratory, invasive, and metastatic ability of RCC cells, and METTL14, YTHDF2 overexpression suppressed tumor growth and metastasis (*Liu et al., 2022d*). On the contrary, depletion of METTL3 could significantly decrease ccRCC tumor cell growth (*Zhu et al., 2022*). Knockdown of METTL3 (*Shi et al., 2021*) or WTAP (*He et al., 2021*) decreased RCC migration and proliferation. Cell proliferation in RCC was also suppressed by knockout of WTAP or IGF2BP (*Ying et al., 2021*). The function of FTO is ambiguous. Some found that FTO knockdown inhibited cell migration and proliferation (*Shen et al., 2022*), and FTO negatively influenced autophagy

(*Xu et al., 2022b*). But it was also be validated that the proliferation of ccRCC cells was suppressed by higher expression of FTO (*Zhuang et al., 2019*). Above all, m6A RNA methylation regulators play an important role in the development of RCC. However, the changes in m6A modulators in different types of RCC are not the same. These need more research to clarify the function of m6A in renal cell carcinoma and explore its change and mechanism.

## Bladder cancer

Bladder cancer is the ninth most common tumor among all types of tumors worldwide, with about 570,000 new cases per year (*Sung et al., 2021*). More and more studies have shown that bladder cancer was related to m6A modification. The m6A modification content in bladder cancer tissues was lower than that in normal tissues (*Gu et al., 2019*), but some studies also found that the m6A modification level was increased in bladder cancer samples (*Liu et al., 2022a*), which may be related to the different sample source and small sample size. It was found in humans and animal models that METTL3 (*Cheng et al., 2019*; *Jin et al., 2019*; *Xie et al., 2020*), FTO (*Tao et al., 2021*; *Zhou et al., 2021*), IGF2BP1 (*Xie et al., 2021*), IGF2BP3, YTHDF1 (*Zhu et al., 2023*), YTHDF 2, ELAV-like protein 1 (ELAVL1), HNRNPA2B1 (*Deng et al., 2022*) were upregulated in bladder cancer cells. In contrast, METTL14 (*Gu et al., 2019*; *Guimarães Teixeira et al., 2022*), WTAP (*Liu et al., 2021a*), YTHDC1, YTHDF3, ZC3H13 (*Deng et al., 2022*) were reduced in bladder cancer samples. However, it was also reported that METTL14 mRNA transcription level was significantly increased (*Liu et al., 2022a*). The change of the expression level of m6A regulators could affect cell function. Stable knockdown of METTL3 (*Cheng et al., 2019*; *Yang et al., 2019*; *Han et al., 2019*; *Xie et al., 2020*; *Wang et al., 2021a*) and IGF2BP1 (*Xie et al., 2021*) effectively reduced cell cycle process, cell proliferation, migration, and invasion ability, while inducing apoptosis and carcinogenic transformation. Knockout of METTL14 enhanced invasion ability and tumor proliferation ability (*Gu et al., 2019*). Using METTL14 knockout cells resulted in smaller tumor size and fewer tumor surrounding blood vessels (*Guimarães Teixeira et al., 2022*). However, there are also other studies showing that ectopic expression of METTL14 increased tumor cell survival ability, proliferation and migration ability (*Liu et al., 2022a*). After knocking down METTL14, cell apoptosis increased, migration and proliferation decreased instead (*Guimarães Teixeira et al., 2022*). FTO was closely related to the self-renewal of bladder tumor stem cells (*Gao et al., 2020*). FTO inhibited the growth of bladder cancer cells by G1 phase arrest and inhibits their migration ability (*Sun et al., 2022*). But at the same time other studies demonstrated that, after FTO overexpression, cells in G0/G1 phase were less while those in S phase increased, cell proliferation ability migration and invasion ability all increased (*Zhou et al., 2021*). Depletion of YTHDF2 reduced Epithelial-mesenchymal transition (EMT) pathway-related proteins and matrix metalloproteinase family proteins expression so that reduced cell migration rate (*Xie et al., 2020*). Different m6A regulators showed different functions in bladder cancer. There are many references found the correlation of bladder cancer and writers and erasers. Further research related to readers is still needed to clarify the role of each m6A regulator.
**Table 1  The variation of the level of m6A regulators in RCC.**

| Function | M6A regulators | Sample | Detection method | Variation | References |
|---|---|---|---|---|---|
| Writer | METTL3 | 84 pairs of RCC and matched normal tissues purchased from Shanghai Outdo Biotech Co., Ltd. | IHC | ↑ | *Zhu et al. (2022)* |
| | | 16 paired ccRCC samples and matched adjacent normal kidney tissues from Shengjing Hospital of China Medical University | qRT-PCR | ↑ | *Zhao, Tao & Chen (2020)* |
| | METTL14 | TCGA, ICGC, Ruijin-RCC dataset | Bioinformatics analysis, Western blots, IHC staining | ↓ | *Zhang et al. (2021a)* |
| | | TCGA, A total of 210 RCC tissues and 47 pairs of RCC and matched normal tissues from the tissue sample database of Changzheng Hospital | Bioinformatics analysis, RT-qPCR , western blot, IHC | ↓ | *Liu et al. (2022d)* |
| | WTAP | TCGA, Oncomine database, 24 paired tissues collected at the First Affiliated Hospital of Medical College, Zhejiang University | Bioinformatics analysis, RT-qPCR | ↑ | *Ying et al. (2021)* |
| | | TCGA, UALCAN database, HK-2, 786-O, and Caki-1 cell lines were obtained from the Cell Bank of the Chinese Academy of Sciences | Bioinformatics analysis, qRT-PCR, Western blots | ↑ | *He et al. (2021)* |
| | EIF3A | TCGA, 30 paired ccRCC tissues and normal renal tissues were purchased from BioChip | Bioinformatics analysis, IHC | ↓ | *Zhang et al. (2021b)* |
| Eraser | FTO | TCGA, GEO, CPTAC | Bioinformatics analysis | ↑ | *Xiao et al. (2020)* |
| | | TCGA, Oncomine, UALCAN, GEO, HK-2, 786-O, Caki-1 and ACHN were purchased from the Cell Bank of the Chinese Academy of Sciences, 24 paired specimens from the First Affiliated Hospital of Medical College, Zhejiang University | Bioinformatics analysis, RT-qPCR, Western blot | ↑ | *Shen et al. (2022)* |
| | | 147 ccRCC, 31 pRCC, 10 chRCC, 13 sarcomatoid renal cell carcinoma, 10 oncocytoma and 30 normal renal tissues from the Biobank at the CIO Cologne-Bonn | IHC | ↓ | *Strick et al. (2020)* |
| | | TCGA, 35 pairs of primary ccRCC and adjacent normal tissues from Peking University Shenzhen Hospital | Bioinformatics analysis, RT-qPCR, Western blot, IHC | ↓ | *Zhuang et al. (2019)* |
| | ALKBH5 | 166 ccRCC and 106 normal renal tissues, 147 ccRCC, 31 pRCC, 10 chRCC, 13 sarcomatoid renal cell carcinoma, 10 oncocytoma and 30 normal renal tissues from the Biobank at the CIO Cologne-Bonn | RT-qPCR, IHC | ↓ | *Strick et al. (2020)* |

**Table 1** (*continued*)

| Function | M6A regulators | Sample | Detection method | Variation | References |
|---|---|---|---|---|---|
| | | TCGA | Bioinformatics analysis | ↓ | *Xu et al. (2022a)* and *Shen et al. (2022)* |
| | YTHDF2 | TCGA | Bioinformatics analysis | ↓ | *Su et al. (2021)* |
| | HNRNPA2B1, YTHDC1, YTHDF1, and YTHDF3 | 166 ccRCC and 106 normal renal tissues, 147 ccRCC, 31 pRCC, 10 chRCC, 13 sarcomatoid renal cell carcinoma, 10 oncocytoma and 30 normal renal tissues from the Biobank of the Center for Integrated Oncology Cologne-Bonn | RT-qPCR, IHC | ↓ | *Von Hagen et al. (2021)* |
| Reader | IGF2BPs | Oncomine database, 24 paired specimens at the First Affiliated Hospital of Medical College, Zhejiang University | Bioinformatics analysis, RT-qPCR | ↑ | *Ying et al. (2021)* |
| | IGF2BP2 | TCGA, GEPIA database, 5 pairs of matched RCC tissues and adjacent normal tissues from the Urology Department of Peking University First Hospital | Bioinformatics analysis, RNA-seq | ↓ | *Li et al. (2022)* |

**Notes.**

RCC, Renal cell carcinoma; pRCC, Papillary renal cell carcinoma; chRCC, Chromophobe renal cell carcinoma; ICGC, International Cancer Genome Consortium; IHC, Immunohistochemistry.

## Prostate cancer

Prostate cancer, as one of the six most common tumors in the world (*Soerjomataram & Bray, 2021*), has also been proven to be closely related to m6A. In prostate cancer, m6A modification level was upregulated, and the expression levels of METTL3 (*Chen et al., 2021b*), METTL14 (*Barros-Silva et al., 2020*), VIRMA (*Barros-Silva et al., 2020*), RNA binding motif protein 15 (RBM15), HNRNPC, HNRNPA2B1, YTHDC2, YTHDF1 (*Liu et al., 2022c*) and YTHDF2 (*Li et al., 2020*) increased. While FTO (*Zou et al., 2022*; *Zhu, Li & Xu, 2021*), ALKBH5 (*Wu et al., 2021a*), ZC3H13 (*Zhang et al., 2021c*) and IGF2BP2 (*Liu et al., 2022c*) decreased. The expression of METTL3 and METTL14 also significantly increased in castration-resistant prostate cancer (*Li et al., 2023b*). A special case is IGFBP3, which had a lower expression level in prostate cancer tissues, but a higher expression level in patients with advanced stage (*Quan, Zhang & Ping, 2022*). Then the rule found out and summarized from the experiment is that prostate cancer was positively related to methyltransferases and negatively related to demethylases. Knockdown of METTL3 reduced the proliferation, migration, invasion and DNA synthesis ability of prostate cancer cells, and inhibited tumor formation and reduced lung metastasis (*Chen et al., 2021b*; *Mao et al., 2022*). This ability was produced by catalyzing m6A modification (*Yuan et al., 2020*). Overexpression of METTL14 positively regulated the number of cells in S phase and negatively regulated the number of cells in G2 phase to improve cell proliferation ability (*Wang et al., 2022*). Upregulated FTO (*Zhu, Li & Xu, 2021*; *Zou et al., 2022*) or knockdown of VIRMA (*Barros-Silva et al., 2020*) inhibited cell proliferation and invasion ability. But there are also studies showing that knockdown of FTO reduced the migration and invasion

ability of prostate cancer cells (*Su, Wang & Li, 2021*). Overexpression of YTHDF2 can also promote cell proliferation and migration ability (*Li et al., 2020*).

To sum up, urogenital cancer had no specific relation with the increase or decrease of m6A modification. There are still some researches demonstrating opposite results. However, the change of m6A modification and regulators' expression could alter the biological behaviors of tumor cells. The mechanism of these changes needs further research in our work from now on.

## Mechanism
### Regulating RNA stability

M6A modification has various functions in cells. One of the essential assignments is to regulate mRNA expression levels and affect the stability of different RNAs. This feature may also be one of the mechanisms by which m6A regulators play a role in the urinary system. The writer and eraser regulate mRNA stability by regulating m6A modification levels, while the reader recognizes m6A modified regions, interacts with mRNA, and affects its decay, regulating its stability. The m6A modification catalyzed by METTL3 in SETD7, KLF4 (*Xie et al., 2020*), LHPP and NKX3–1 (*Li et al., 2020*) was recognized by YTHDF2 to promote mRNA degradation. It was also reported that the decay of METTL14-methylated bromodomain PHD finger transcription factor (BPTF) (*Zhang et al., 2021a*) and THBS1 (*Wang et al., 2022*) accelerated. Eraser FTO promoted the demethylation of PPARg coactivator-1 $\alpha$ (PGC-1 $\alpha$) (*Zhuang et al., 2019*) and CLIC4 (*Zou et al., 2022*) and enhanced mRNA stability. But there are also other researches demonstrated that the m6A modification mediated by METTL3, METTL14 (*Zheng et al., 2022*) and WTAP (*Ying et al., 2021*) maintained the stability of mRNA. IGF2BPs stabilized their target genes by the read of m6A modified sites (*Gu et al., 2021*; *Ying et al., 2021*; *Li et al., 2022*; *Huo et al., 2022*). Meanwhile, m6A regulators modulated the interaction between transcripts and other proteins to affect the stability of mRNA. METTL3 knockdown regulated the interaction between Rho GDP dissociation inhibitor $\alpha$ (ARHGDIA) and ELAV-like RNA-binding protein 1 (ELAV1) and the interaction between ELAV1 and ubiquitin-specific protease 4 (USP4) to impact its function (*Chen et al., 2021b*). But the mechanism behind this phenomenon is not known. We summarized the mRNA and its functional role which related to m6A modification in Table 2. Others kind of RNA also modulated mRNA stability *via* m6A modification. A Circular RNA circPTPRA bound with IGF2BP1, blocking its recognition of downstream m6A modified mRNA and reducing target mRNA stability (*Xie et al., 2021*; *Zheng et al., 2023*). Conversely, LncRNA TRAF3IP2 antisense RNA 1 (TRAF3IP2-AS1) accelerated the poly ADP-ribose polymerase (PARP1) mRNA decay by stimulating m6A mRNA modification (*Yang et al., 2021a*). Taken together, although the function of m6A modification in various mRNA stability was different, there is no doubt that regulating RNA stability is one of the essential functions of m6A modification.

### Regulating gene expression

The mRNAs with different degrees of m6A methylation have different effects on tumor occurrence and development. M6A regulators and other enzymes regulated gene expression

**Table 2  The adjustment of mRNA stability *via* m6A modification and related m6A regulators.**

| mRNA | Role in urological cancers | Related m6A regulators and its function | Impact on mRNA | Reference |
|---|---|---|---|---|
| SETD7 | Suppressor | METTL3: Catalysed m6A modification; YTHDF2: Recognized modified m6A site to mediate the mRNA decay | Degradation | *Xie et al. (2020)* |
| KLF4 | Suppressor | METTL3: Catalysed m6A modification; YTHDF2: Recognized modified m6A site to mediate the mRNA decay | Degradation | *Xie et al. (2020)* |
| LHPP | Suppressor | METTL3: Upregulated the level of m6A modification; YTHDF2: Recognized modified m6A site to mediate the mRNA decay | Degradation | *Li et al. (2020)* |
| NKX3–1 | Suppressor | METTL3: Upregulated the level of m6A modification; YTHDF2: Recognized modified m6A site to mediate the mRNA decay | Degradation | *Li et al. (2020)* |
| ARHGDIA | NA | METTL3: The knock down of METTL3 elevated decay and enhanced the interaction between ELAVL1 and ARHGDIA | Degradation | *Chen et al. (2021b)* |
| ELAVL1 | NA | METTL3: The knock down of METTL3 enhanced the interaction between USP4 and ELAVL1 to decrease ubiquitination level of ELAVL1 | Degradation | *Chen et al. (2021b)* |
| USP4 | NA | METTL3: Catalysed m6A modification to promote the binding of HNRNPD to USP4 | Degradation | *Chen et al. (2021b)* |
| DBT | Suppressor | METTL3: Catalysed m6A modification and decreased half-life of mRNA. | Degradation | *Miao et al. (2023)* |
| NAP1L2 | Promoter | METTL3/METTL14: Catalysed m6A modification; HNRNPC: Maintain the stability of NAP1L2 mRNA | Stability | *Zheng et al. (2022)* |
| TFAP2C | NA | METTL3: Promoted m6A methylation; IGF2BP1: recognized the m6A site to enhances TFAP2C mRNA stability | Stability | *Wei et al. (2020)* |
| S1PR3 | Promoter | WTAP: Bound to the S1PR3 to Catalysed m6A modification; IGF2BP: Bound to the S1PR3 to maintain the stability | Stability | *Ying et al. (2021)* |
| PGC-1$\alpha$ | Suppressor | FTO: Catalysed demethylation | Stability | *Zhuang et al. (2019)* |
| CLIC4 | NA | FTO: Catalysed demethylation | Stability | *Zou et al. (2022)* |
| ZNF677 | Suppressor | METTL3: Catalysed m6A modification to prolong the half-life and elevate the translation of ZNF677; IGF2BP2: Bound to the modified m6A site to increase the mRNA stability; YTHDF1: Participated in the translation ZNF677 | Stability | *Li et al. (2022)* |
| PPAT | Promoter | IGF2BP2: Interact with PPAT mRNA to promote the stability of PPAT transcripts | Stability | *Huo et al. (2022)* |
| CDK4 | Promoter | IGF2BP3: Cooperated with DMDRMR to read modified m6A site and enhance mRNA stability | Stability | *Gu et al. (2021)* |
| THBS1 | Suppressor | METTL14: Catalysed m6A modification; YTHDF2: | Degradation | *Wang et al. (2022)* |
| PARP1 | Promoter | NA | Degradation | *Yang et al. (2021a)* |

**Notes.**
SETD7, SET domain containing 7; KLF4, Kruppel-like factor 4; ARHGDIA, Rho GDP dissociation inhibitor $\alpha$; ELAVL1, ELAV-like RNA-binding protein 1; USP4, Ubiquitin specific protease 4; DBT, dihydrolipoamide branched chain transacylase E2; HNRNPD, Hetergeneous nuclear ribonucleoprotein D; TFAP2C, Transcription factor-activating enhancer-binding protein 2C; PGC-1$\alpha$, PPARg coactivator-1 $\alpha$; ZNF677, Zinc finger protein 677; PPAT, Phosphoribosyl pyrophosphate amidotransferase; CDK4, Cyclin-dependent kinase 4; DMDRMR, DNA methylation–deregulated and RNA m6A reader–cooperating lncRNA; THBS1, Thrombospondin 1; PARP1, Poly ADP-ribose polymerase.
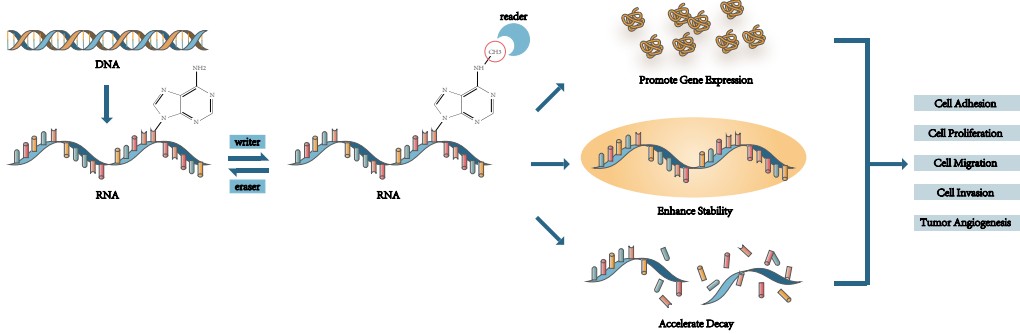

**Figure 1** **The mechanism of m6A affecting the cell phenotype and mediating cell functions.** M6A regulators regulate the level of m6A modification and recognize the site of m6A to change the cell phenotype and function.

by changing the m6A methylation level of mRNA, thereby influencing cell function. In RCC, methylenetetrahydrofolate dehydrogenase 2 (MTHFD2) regulated hypoxia-inducible factor (HIF)-2$\alpha$ mRNA methylation dependent on METTL3, thereby increasing HIF-2$\alpha$ protein levels and promoting glycolysis (*Green et al., 2019*). CUB-domain containing protein 1 (CDCP1) is a kind of transmembrane glycoprotein which associated with metabolic pathways (*Khan et al., 2021*). METTL3 and YHDF1 bound to CDCP1, promoting CDCP1 mRNA modification and translation (*Yang et al., 2019*). METTL3 promoted m6A modification in PCAT6 and increased PCAT6 expression in an IGF2BP2-dependent manner (*Lang et al., 2021*). M6A modification, which induced be METTL14, inhibited the expression of oncogene nuclear-enriched abundant transcript 1_1 (NEAT1_1) in RCC through YTHDF2-dependent RNA degradation (*Liu et al., 2022d*). SOD2, a superoxide dismutases, was positively correlated with WTAP expression and WTAP affected the expression of SOD2 by regulating SOD2's m6A methylation level (*Liu et al., 2021a*). In bladder cancer, the upregulation of circSLC38A1 was associated with m6A modification, circSLC38A1 promoted tumor invasion and metastasis (*Li et al., 2023a*). M6A modification could also regulate the distribution of Circular RNA in cells. CircMET was exported to the cytoplasm by YTHDC1 through m6A modification. When circMET was distributed in the cytoplasm, cell proliferation and growth were inhibited (*Yang et al., 2022*).

In a word, m6A regulators modulated downstream gene expression to regulate cell metabolism, and the mechanism behind was concluded in Fig. 1. It is a worthy research task how to control tumor occurrence and development in this way.

## Modulating the immune system

It is increasingly recognized that m6A RNA methylation regulators are closely related to the human immune system. M6A score was used to quantify m6A modification patterns, which could predict the immune phenotype of RCC patients (*Li et al., 2021b*). The tumour microenvironment (TME) is one of the important factors that influence the progress and oncogenesis. The infiltrating immune cell formed a stimulating or inhibiting network to transform TME (*Li et al., 2021c*). People judged the degree of immune cell infiltration

in cancer tissues by detecting m6A modification level and the expression level of m6A regulators. The expression level of m6A RNA methylation regulators affected the immune infiltration of urogenital tumors. The level of m6A modification was related to T cell infiltration degree (*Liang et al., 2022*). In RCC, a study grouped according to 23 m6A regulators found that different groups had different degrees of immune infiltration (*Deng et al., 2022*). High EIF3A expression upregulated the overall immune infiltration level (*Zhang et al., 2021b*). METTL14 inhibited Tregs infiltration in ccRCC by inhibiting CCL5 (*Xu et al., 2021a*). The expression of IGF2BP3 was demonstrated to be closely related to the infiltration of immune cells in PRCC (*Chen et al., 2021a*), which was mainly related to Th2 cell infiltration enrichment and affected PD-1 treatment effect (*Zhong et al., 2021*). In bladder cancer, cluster analysis based on the expression level of m6A regulators found that nearly all of immune cells in the m6ACluster B group decreased. Patients in the m6ACluster B group had a better prognosis than others (*Ye et al., 2021*). Also, the B group in another research was closely related to immune activation (*Zhu et al., 2021*).

M6A RNA methylation regulators were closely related to immune cells. It was validated that m6A regulators partook in the differentiation and recruitment process of immune cells (*Deng et al., 2022*). Changes in METTL3 expression were closely associated to the down-regulation of CD4 + T cells, neutrophils, and dendritic cells infiltration level in bladder cancer (*Wang et al., 2021b*), while positively correlated with naive B cells, CD4+ memory resting T cells, and M1 macrophages number and negatively correlated with regulatory T cells and M2 macrophages number in prostate cancer (*Liu et al., 2022c*). The expression of YTHDF2 was positively correlated with B cells, CD8+T cells, CD4+T cells, macrophages, neutrophils, and dendritic cells. It was be validated that this correlation influenced treatment outcomes (*Su et al., 2021*). FMR1 expression was significantly negatively associated with M2 macrophages, while decreased M0 macrophages in urothelial carcinoma tissues upon HNRNPA2B1 overexpression (*Kong et al., 2022*). In ccRCC, *Chen et al. (2020b)* confirmed that a higher m6A risk score leaded to a higher abundance of regulatory T cells (Tregs) and follicular helper T cells (Tfhs). On the contrary, in bladder cancer, CD56dim natural killer cells, central memory CD4T cells, eosinophils, mast cells, monocytes, and17-type helper T cell numbers were lower in high-m6Ascore group, while activated CD4 + T cell, memory B cell, natural killer T cell, neutrophil, 2-type helper T cell *etc.*, and mainly related to immunosuppression-related cell numbers were less in the low-m6Ascore group (*Deng et al., 2022*). However, another study of bladder cancer demonstrated that the m6A score was positively correlated with CD4 T immune cell, CD8 T immune cell, and dendritic immune cell number, while negatively correlated with PD-L1 expression level (*Zhu et al., 2021*). At the same time, studies also demonstrated that the number of each group's immune cells was different through cluster analysis (*Liu et al., 2021b*).

After a hundred years of development, immunotherapy has also become one of the important treatment options for tumor patients. Anti-PD-1 immunotherapy as an emerging first-line treatment option has been used for treatment of advanced ccRCC (*Atkins & Tannir, 2018*) and it is also a promising treatment option for patients with advanced bladder cancer (*Inman et al., 2017*). Several studies indicated that IGF2BP3 was positively associated with the expression of PD-L1 in immune cells (*Cui et al., 2022*), while

FTO was just the opposite (*Deng et al., 2022*). *Kong et al. (2022)* found that in bladder cancer, the expression of HNRNPA2B1, FMR1, IGF2BP1, IGF2BP3, and YTHDF2 were significantly upregulated in the anti-PD-1 immunotherapy response group, while FTO expression was significantly downregulated. They also constructed a nomogram to predict the responsiveness of patients to atezolizumab monotherapy (*Kong et al., 2022*). But there were also other studies showing that FTO expression level was higher in bladder cancer patients who did not respond to immunotherapy (*Deng et al., 2022*). In addition, patients with low m6A score had a better response to immunotherapy. The influence of m6A modification and m6A regulators on the immune system provides possibilities for discovering new treatment methods, selecting suitable treatment methods for different patients, and performing precision medicine in the future.

## Clinical value
### The prognostic value
M6A regulators are closely related to the malignant progression of tumors, and numerous studies have shown that they can be used to predict clinical outcomes. Increasing evidence demonstrated that patients with different expressions of m6A regulators had dissimilar prognoses. The same m6A regulator has also played different role in various urinary system tumors. In RCC, low expression of METTL14 (*Zhang et al., 2021a*; *Liu et al., 2022d*), EIF3A (*Zhang et al., 2021b*), FTO (*Zhuang et al., 2019*), ALKBH5 (*Strick et al., 2020*), YTHDF2 (*Mu et al., 2020*; *Su et al., 2021*; *Xu et al., 2022a*), ZC3H13, and KIAA1429 (*Wang et al., 2020*) were associated with poor prognosis, while low levels of METTL3 (*Xu et al., 2021a*; *Zhu et al., 2022*), WTAP (*He et al., 2021*), IGF2BPs (*Ying et al., 2021*), YTHDC1, YTHDF1, and YTHDF3 (*Von Hagen et al., 2021*) were associated with longer overall survival of patients. Higher IGF2BP3 and HNRNPC lead to shorter survival time in PRCC (*Chen et al., 2021a*), and an increased FTO expression upregulated overall survival rate in pRCC patients (*Guimarães Teixeira et al., 2021*). In bladder cancer, it has demonstrated that ALKBH5, IGFBP2-3, RBM15, RBMX, YTHDC1, and YTHDF had prognostic value (*Liu et al., 2021b*). METTL14 (*Li et al., 2021a*), YTHDC1, and WTAP (*Cui et al., 2022*) may be protective factors for bladder cancer, the higher expressions were related to the longer overall survival of patients, while IGF2BP1-3 (*Xie et al., 2021*), YTHDF1 (*Zhu et al., 2023*), LRPPRC (*Cui et al., 2022*), ALKBH5, and FTO (*Deng et al., 2022*) were risk factors for bladder cancer. The higher the FTO expression level, the lower the overall survival rate (*Tao et al., 2021*) and the shorter disease-free survival period (*Gao et al., 2020*, p. 42020). In prostate cancer, the higher the expression of METTL3 and YTHDF2, the lower the survival rate of prostate cancer patients (*Li et al., 2020*). The level of VIRMA was negatively associated with the patient's disease-free survival rate (*Barros-Silva et al., 2020*). Patients with high METTL14 expression had shorter OS (*Wang et al., 2022*), while low FTO expression was associated with poor prognosis (*Zou et al., 2022*). After further processing of m6A regulator expression levels, it has demonstrated that using METTL3 and METTL14 to formulate risk characteristic was negatively correlated with prognostic characteristics (*Chen et al., 2020a*). The risk score, established by METTL14, HNRNPA2B1, and YTHDF2, was also negatively correlated with prognosis (*Liu et al., 2022c*). The lower

m6A score, calculated by m6A regulator expression levels, the worse prognosis for bladder cancer (*Zhu et al., 2021*) and testicular germ cell tumors (TGCT) (*Cong et al., 2021*) patients. The prostate cancer patients in the m6A high score group had a worse prognosis than the m6A low score group (*Liu et al., 2022b*). In addition, METTL3's copy number variations (CNVs) were related to OS (*Wang et al., 2021b*). Various m6A-related LncRNA were also significantly related to RCC patients' prognostic risk (*Xia et al., 2022*).

M6A regulators were closely related to tumor occurrence and development and had a significant correlation to the clinical characteristics of patients. Through research and analyses, it was found that METTL14 level was negatively linked with tumor size, metastasis stage, pathological grade, and TNM stage (*Zhang et al., 2021a*; *Liu et al., 2022d*), and METTL14 could increase lymph node metastasis, liver metastasis and mortality rate of BC mouse model (*Liu et al., 2022a*). IGF2BP1 was positively correlated with tumor size, lymph node metastasis, and late clinical-stage of bladder cancer (*Xie et al., 2021*). Low expression of HNRNPC, METTL3, YTHDF2, and RBM15 could be found in high-grade tissue samples (*Wang et al., 2020*). On the contrary, FTO and YTHDF2 were proved that the more they contained in tissues, the higher the tumor stage, and the worse the prognosis (*Von Hagen et al., 2021*; *Zhou et al., 2021*). In addition, the expression levels of METTL3 (*Li et al., 2020*), FTO (*Zhu, Li & Xu, 2021*), YTHDF1-2, YTHDC2 (*Wu et al., 2021a*) were positively correlated with the Gleason score of prostate cancer. The m6A score could be used to predict the tumor grade of bladder cancer patients. *Liu et al. (2021b)* reported that patients with higher m6A scores had higher tumor stage and worse prognosis. The lower m6A score prostate cancer patients had higher rate of tumor metastasis and recurrence (*Quan, Zhang & Ping, 2022*). It can be seen that m6A as a current research hotspot has the potential in predicting the prognosis of tumor patients. The next working direction is how to apply in clinical diagnosis and prognostic evaluation.

### Target in therapies

M6A methylation is a promising therapeutic research field. Researchers have disclosed that m6A regulators play key roles in cancer treatment and are potential therapeutic targets worth exploring. It was demonstrated that existing treatments were closely related to m6A. Sunitinib is a molecular targeted drug used to treat advanced RCC, and studies showed that, in sunitinib-resistant RCC, METTL14-mediated m6A modification significantly enhanced the stability of tumor necrosis factor receptor-associated factor 1 (TRAF1), and overexpression of TRAF1 increased sunitinib resistance (*Chen et al., 2022*). This experiment suggests that we may be able to reduce the patient's resistance to sunitinib by regulating the expression level of METTL14. In patients treated with cisplatin for bladder cancer, it has demonstrated that cisplatin regulated METTL3's m6A methylation function and reduced the expression level of granulocyte colony-stimulating factors (G-CSF) mRNA, and inhibited the expansion and immune suppression ability of fibrocytic myeloid-derived suppressor cells (f-MDSCs) (*Mu et al., 2021*). In TGCT, knockdown of VIRMA enhances sensitivity to cisplatin *in vivo* (*Miranda-Gonçalves et al., 2021*). Endocrine therapy is an important means of treating prostate cancer patients, which works by reducing or eliminating the effect of hormones on tumors. In prostate cancer, the expression levels of

METTL14, FTO, YTHDC1-2, and YTHDF1-3 were positively correlated with androgen receptor expression (*Wu et al., 2021a*), suggesting that we may be able to assist in the treatment of prostate cancer by regulating the expression levels of m6A regulators so that increasing patient sensitivity to castration therapy and enhancing drug efficacy. At the same time, m6A was closely related to how much patients can benefit from immunotherapy. In the clinical treatment process, it has demonstrated that patients with high m6A scores could obtain more prolific from immune checkpoint therapy (ICT) (*Xu et al., 2021b*). In the low m6A score group, the sensitivity of targeted therapy drugs decreased while the clinical benefit rate of anti-PD-1 treatment rose (*Li et al., 2021b*). It is high m6Ascore group responded better to PD-1 treatment in bladder cancer (*Deng et al., 2022*). When patients used anti-PD1 and anti-CTLA4 immunotherapy treatment, the higher m6A score group got more survival benefits than others. And when patients used anti-CTLA4 immunotherapy alone, the low m6A score group had better efficacy (*Liu et al., 2021b*). MRPRS is established by using eight m6A RNA methylation regulators and is considered a promising biomarker. MRPRS was negatively correlated with patient survival outcomes. Immunotherapy was more suitable f High MRPRS patients while targeted therapy was better for low MRPRS patients (*Yu et al., 2022*). These results indicate that m6A regulators are closely related to the treatment effect of tumor patients. m6A score and MRPRS may be potential predictive factors for determining treatment methods. Through the efforts of researchers, relevant inhibitors have also been invented, providing the possibility for future treatment. UZH2 is a newly synthesized small molecule METTL3 inhibitor that could reduce the level of m6A modification in prostate cancer cells (*Dolbois et al., 2021*). Mice treated by the FTO small-molecule inhibitors FB23-2 had smaller tumor size and longer survival time (*Xu et al., 2022b*). We are looking forward to further clinical research about oncotherapy and m6A modification.

## CONCLUSIONS

Nowadays, it is increasingly recognized m6A modification plays an important role in tumors of the urinary system. There is considerable focus on exploring its possible mechanisms of action and looking for its clinical application value. This review mainly summarized the research on m6A modification and kidney cancer, bladder cancer, and prostate cancer so far, analyzed the changes of m6A modification and m6A regulators in cells and found that m6A modification could regulate RNA stability, control RNA decay, change gene expression levels, interact with the immune system, thereby regulating cell proliferation, migration and invasion, changing cell phenotype, affecting tumor occurrence and development. Many studies showed that m6A had the potential to predict patient prognosis and became a director for researching new treatment methods. M6A as a research hotspot had important significance in the study of urological tumors. In our research we also found that there are still many contradictions and questions waiting for the answers: (1) Based on different sample data, the results of experiments were different. Why do these studies produce opposite results? Is the level of m6A positively or negatively correlated with RCC? What is the role of FTO in promoting or inhibiting cell division, proliferation and invasion in

urinary system tumors, and what role does it play in tumor genesis and development? (2)The expression of the same m6A regulator is different in different pathological types of RCC. What is the connection between the pathological types and m6A modification? (3) The effect of m6A modification on the stability of different RNAs varied. What is the underlying mechanism behind it? (4) We have found a close relationship between m6A modification and urinary system tumors. Further research is still needed to find appropriate and enabling m6A related indicators that can be applied to make clinical treatment strategies and predict prognosis of patients.

### Funding
This study was supported by the Regional Science Foundation Project approval number (82160544). The funders had no role in study design, data collection and analysis, decision to publish, or preparation of the manuscript.

### Grant Disclosures
The following grant information was disclosed by the authors:
Regional Science Foundation Project: 82160544.

### Competing Interests
The authors declare there are no competing interests.

### Author Contributions
- Yaohui Zeng conceived and designed the experiments, performed the experiments, analyzed the data, prepared figures and/or tables, authored or reviewed drafts of the article, and approved the final draft.
- Cai Lv conceived and designed the experiments, performed the experiments, analyzed the data, prepared figures and/or tables, authored or reviewed drafts of the article, and approved the final draft.
- Bangbei Wan performed the experiments, analyzed the data, authored or reviewed drafts of the article, and approved the final draft.
- Binghao Gong performed the experiments, analyzed the data, authored or reviewed drafts of the article, and approved the final draft.

### Data Availability
This is a literature review.

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
