# Peer review of "The current landscape of m6A modification in urological cancers"

_PeerJ, doi:10.7717/peerj.16023_

## Round 0.1 · original submission · Major Revisions

Please attend to the comments and resubmit at the earliest.

Reviewer 1 ·

Basic reporting

- The topic was reviewed in 2020 by Yang et al. (PMID: 33015074). It would be beneficial to clearly articulate the motivation for this new review and highlight the additional information or insights offered compared to the previous review. Although Yang et al. reviewed similar topics in 2020, the authors of this manuscript made extensive references to reports from more recent literature.
- Some scientific language edits are recommended to enhance clarity and readability.
- For short lists like in lines 30-33: "We found that RNA modification was associated with embryonic development, cardiovascular diseases, mitochondrial diseases, neurological disorders, and human cancer, especially human cancer (Barbieri & Kouzarides, 2020; Suzuki, 2021; Wu et al., 2021b; Song et al., 2021)." Instead of having the references all compiled at the end – if possible, it could be helpful if the relevant references were placed next to the specific claims or findings they support. This would allow readers to find appropriate references more efficiently.
- Line 34 – please reconsider the phrasing "over 300 RNA modification methods". The term "methods" could be misinterpreted as experimental techniques for modifying RNA. Instead, "RNA modifications" or "types of post-transcriptional modifications" would more accurately represent naturally occurring biological processes.
- Line 36-37 "The most commonly occurring and the hottest research field is m6A" – this claim is subjective. Please consider modifying this claim to be more objective or provide some appropriate evidence or references for support.
- Line 39-40: "We texted the genome-distribution of m6A in 2022" – it wasn't clear what this sentence is describing.
- Line 48-56 are missing appropriate references for the descriptions of "Erasers", "Readers", etc.
- Line 57: "All sorts of biological processes were regulated by m6A modification." – this statement is somewhat vague. I suggest directly specifying the particular biological processes that are most significantly affected or regulated by m6A modification.
- "They acted on RNA splicing, nuclear transfer, stability, degradation, translation, and regulate the interaction between RNA and protein (Wang et al., 2015; Lin et al., 2016; Deng et al., 2018; Frye et al., 2018)" – again, place references next to the specific claims or findings they support.
- Lines 62-68 – the paragraph lacks references.
- Table 1: it may help enhance readability to avoid repeating the "Functions" column and protein names. The same functions and proteins could be grouped together with multiple entries for different sources.
- Line 178 – please correct the spelling of "Mechanism."
- In the "Modulating the immune system" section starting on line 239, please ensure to provide sufficient context and background for the immune processes described. It's crucial that these processes are depicted accurately. For instance, the terms "immune infiltration" and "immune response" describes different events and should be used appropriately based on context.
- Line 359 – "we found" doesn't appear appropriate unless the authors were also involved in the cited work.
- The conclusion, as currently written, appears somewhat vague. It could be improved by offering more detailed and insightful discussions and proposals for future directions.

Experimental design

- Line 76-77: "We used the keywords 'm6A', 'N6-methyladenosine', 'm6A modification', 'm6A regulators', 'urological cancer', 'urological tumor', 'renal cell carcinoma', 'bladder cancer', 'prostate cancer'" – this may miss potentially relevant terms that does not exactly match the keywords, i.e., "urothelial carcinoma", "transitional cell carcinoma", "urethral cancer" etc. A MeSH term search on PubMed with "Urological Neoplasms" and a keyword search with m6A-related keywords may yield more comprehensive and specific results. For example: (Urological Neoplasms[MeSH Terms]) AND ((m6A) OR (N6-methyladenosine) OR (m6A modification) OR (m6A regulators))
- The paragraphs discussing "Renal cell carcinoma" (starting line 84) and "Bladder Cancer" (line 122) are very long. It might be helpful to separate different ideas into different paragraphs within the section. i.e., for the RCC, the paragraphs could be divided into the types of RCC.
- Table 1 summarizes m6A regulators in RCC in an organized way and is easy to understand. The authors could consider doing the same summarization for the other cancer sections. This would help enhance readability.
- For the Mechanism section (beginning on line 178), a summary of the literature in the form of a graphic or table could greatly improve readability and comprehension. This could also help to condense the text, which currently seems to contain many lists of findings. Visual representation of these data could make them more accessible and easier for readers to digest.

Validity of the findings

- While the manuscript seems to have extensively referenced the literature and extracted information from various sources, there appears to be a significant lack of in-depth interpretation and discussion of the findings reported in these studies. Adding more critical analysis and commentary could significantly enhance the value of this review. This could include discussing the implications of each finding, potential controversies/disagreements in the field, and areas where further research is needed.
- While the manuscript describes the levels of various proteins involved in m6A regulation in different urological cancers – there is only minimal introduction to these proteins aside from simply listing them. It would be beneficial to elaborate on the specific roles and functions of these regulators in the relevant sections. This could provide important context and background for further discussions and critical analyses, aiding readers in understanding the significance of these regulators in the context of urological cancers.
- Line 107: "Lei et al. revealed that the level of WTAP reduced in pRCC and chRCC (Lei et al., 2022)" – the study referenced only appears to have described a difference in the transcription level of WTAP in these RCCs. It is important to distinguish the differences between gene expression level differences and protein level differences when referring to these types of findings in the review.
- Line 113: "Either knockout or knockdown of METTL3, WTAP, and IGF2BPs decreased RCC migration and proliferation(Shi et al., 2021; Ying et al., 2021; He et al., 2021)." – please distinguish "and" vs. "or" – i.e., individual knockdown/knockout decreased RCC migration and proliferation or the combination of all three knockdowns.
- I recommend that the authors carefully review the data from the experiments/analysis performed in the referenced literature in the review and provide brief introductions to the specific experiments or analyses performed. This could greatly enhance reader understanding and avoid misinterpretations.
- Line 284-285: "PD-1 monoclonal antibody therapy is the most important immunotherapy option for urological tumors" – please add appropriate reference to support this.

Additional comments

Liu et al. (2022) recently reviewed the relevance of m6A in bladder cancer (PMID: 35087575), which overlaps with some sections of this review. This may be of interest to the authors.

Reviewer 2 ·

Basic reporting

the Introduction adequately introduced the subject of m6A in urologic cancers. I only have a few comments:

1. Line 30. "we found that ..." I think it is better to use "it has been found/reported that..". As these results are not found by the authors

2. Line 39-40 and line359j, same as comment 1#

3. Line 62-68, this paragraph need to cite more literature to support what the author stated

Experimental design

1. for Survey methodology, the authors need to give a certain time frame they conducted this literature search. A review of the recent literature will be preferred to publications from like 10 years ago.

2. though I understand RCC, bladder cancer and prostate cancer are the largest categories in urological cancers, I wonder what is the reason that the authors didn't include other urological cancers like ureters, urethra, testicles, etc.

Validity of the findings

No comment

Additional comments

This review provided a comprehensive overview of the m6A research currently in the field of urologic cancer. The common m6A in kidney, bladder and prostate are discussed relatively and the authors summarized the potential mechanisms of m6A affecting the urologic cancer initiation and progress. Moreover, the clinical and therapeutic significance of m6A was highlighted in the last part.

The review is overall well-written. I only have a few comments and please see above.

Reviewer 3 ·

Basic reporting

The review details current findings on m6A RNA modification in urological cancers. While the review is very detailed and recounts the recent findings, it is poorly written and appears to be a factual dump rather than a review. It is recommended that the authors redesign the overall review structure with sections having clear conclusions and introduction. For example, mechanism section (line 172, which is misspelled) should be moved to the front after introduction to make it more coherent. Line 180 to 185, gave a nice introduction to mechanism but then 185 to 218 is describing one paper per line without any context or line of thought. Only relevant papers should be highlighted with clear rationale while others can be put in table 1. This needs to be repeated for every section to thoroughly revise this review. Additionally, please refrain from using words like 'hottest' to describe a research field.

Experimental design

not applicable

Validity of the findings

not applicable

Additional comments

Manuscript needs to be revised thoroughly and sent for peer review again.

---

## Round 0.2 · Minor Revisions

Please revise and resubmit at the earliest.

Reviewer 1 ·

Basic reporting

The authors have made considerable improvements to the manuscript since the first revision.
Below are some additional comments:

- This comment from the previous round of revision was not addressed: “The topic was reviewed in 2020 by Yang et al. (PMID: 33015074). It would be beneficial to clearly articulate the motivation for this new review and highlight the additional information or insights offered compared to the previous review”.

- There are still some language concerns that may need to be addressed with additional proofreading. Some instances noticed are below:
1. Line 30 “it has demonstrated” (“it has been demonstrated”?)
2. Line 45-46: “It is modulated by kind of m6A RNA methylation regulators.” (it was unclear what “by kind of” means in this context)
3. Line 53: “It is related to…” (“they are related to…”?)
4. Line 60: “A sorts” (“All sorts”?)

- Line 38: “(Dubin & Taylor, 1975)” appears to be a dated reference. Is there any recent literature that corroborates the claim made?

Experimental design

Comments from the first round of review were addressed adequately.
- Please check the reference on line 67, “(Y & H, 2022)”.

Validity of the findings

Comments from the first round of review were addressed. However, I still recommend further improving the quality of this review by adding more novel and insightful analyses and interpretations from relevant literature.

Reviewer 2 ·

Basic reporting

The authors have revised the introduction part according to my comments.

Experimental design

The authors have addressed my questions about the literature they reviewed on and about the rationale of choose certain urological cancer types.

Validity of the findings

No comment

Additional comments

The authors have made revisions according to reviewers' comments and I think now it reaches the requirements of publishing in PeerJ journals.

---

## Round 0.3 · accepted · Accept

Based on the revisions the updated manuscript is improved sufficiently to consider for publication.